# A Spatially Explicit Approach for Targeting Resource-Poor Smallholders to Improve Their Participation in Agribusiness: A Case of Nyando and Vihiga County in Western Kenya

**Mwehe Mathenge [1,\*], Ben G. J. S. Sonneveld [2] and Jacqueline E. W. Broerse [2]**

[1]   Department of Urban Management, School of Planning and Architecture, Maseno University, Private Bag, Maseno 40105, Kenya

[2]   Athena Institute, Faculty of Science, Vrije Universiteit Amsterdam, De Boelelaan 1085, 1081HV Amsterdam, The Netherlands; b.g.j.s.sonneveld@vu.nl (B.G.J.S.S.); j.e.w.broerse@vu.nl (J.E.W.B.)

\*   Correspondence: mmwehe@maseno.ac.ke

**Abstract:** The majority of smallholder farmers in Sub-Saharan Africa face myriad challenges to participating in agribusiness markets. However, how the spatially explicit factors interact to influence household decision choices at the local level is not well understood. This paper's objective is to identify, map, and analyze spatial dependency and heterogeneity in factors that impede poor smallholders from participating in agribusiness markets. Using the researcher-administered survey questionnaires, we collected geo-referenced data from 392 households in Western Kenya. We used three spatial geostatistics methods in Geographic Information System to analyze data—Global Moran's I, Cluster and Outliers Analysis, and geographically weighted regression. Results show that factors impeding smallholder farmers exhibited local spatial autocorrelation that was linked to the local context. We identified distinct local spatial clusters (hot spots and cold spots clusters) that were spatially and statistically significant. Results affirm that spatially explicit factors play a crucial role in influencing the farming decisions of smallholder households. The paper has demonstrated that geospatial analysis using geographically disaggregated data and methods could help in the identification of resource-poor households and neighborhoods. To improve poor smallholders' participation in agribusiness, we recommend policymakers to design spatially targeted interventions that are embedded in the local context and informed by locally expressed needs.

**Keywords:** smallholder farmers; agribusiness; market participation; spatially explicit; GIS; spatial autocorrelation; cluster and outlier analysis; spatial dependency; spatial interventions

## 1. Introduction

Smallholder farmers are important drivers of food security, poverty reduction, and livelihoods in rural and peri-urban areas in developing countries. They produce up to 80% of the food consumed in Sub-Saharan Africa [1,2]. In Kenya for instance, 75% of rural inhabitants are smallholder households who practice smallholder agriculture [3]. However, the majority of smallholder households are disadvantaged in effectively participating in agribusiness activities. Many factors interact to impede their access to and participation in agribusiness markets, including high poverty levels, lack of access to productive resources, and low endowment of human, financial, physical, and socio-economic livelihood capitals, among others [4–9].

The outcome of these factor interactions within individual households is most pronounced at the local level (i.e., farms and neighborhoods). Their spatial manifestation can be observed from the

resulting diverse smallholder farming typologies across rural landscapes [10]. However, how these factors interact spatially to influence smallholder farming decisions is little understood. Deconstructing the local spatial complexity of factors and processes affecting agriculture production could provide a deeper insight into how geographically explicit determinants promote or impede poor smallholder farmers' participation in agribusiness.

The spatial heterogeneity of household livelihood assets endowments has often been used to explain the diversity of smallholder farming typologies and choices across and within geographic locations [11–14]. The different typologies of smallholder farming systems in a given territory can thus be conceptualized as spatial manifestations of individual households' farm management decisions and actions arising from diverse interactions of household livelihood capitals and complex geographic environments [10,13,15–18]. For example, household everyday farming decisions are influenced by the interactions of variabilities of socio-economic, agroecological, biophysical, and institutional variables [4,19–21]. At the lowest spatial unit (farm level), varying biophysical and agroecological constraints (soil variability, water scarcity, topography, pests and diseases, climatic variability, etc.) act as primary determinants of smallholder households agricultural productivity [22]. At a higher spatial unit (territory level), exogenous variables like market structures, transport, technology, off-farm employment, market regulations, etc. interact to influence smallholders' market participation decisions. When socio-economic variables are included into the system (e.g., family size, landholding size, labor, skills, education, and training), a clear spatial variation in the characterization of smallholder farming typologies emerge, adopted to, and distinct from one farm to another and across geographic localities. Hence, the geographic (spatially explicit) variables at the local level, if properly interrogated, could be indispensable in explaining the smallholder farmers' choices to participate, or not, in the higher agribusiness value chains.

According to Głębocki, Kacprzak, and Kossowski [10], spatial dependence is considered a leading effect that influences agriculture practices and decision choices of households. In their study, Głębocki et al. (ibid) was able to map and geo-visualize spatial distribution of smallholder farming typologies by analyzing their spatial dependence characteristics. In the literature, spatial dependence is described as a condition where attribute values are observed at one location depending on the values of neighboring observations at nearby locations [10,23,24]. The assumption taken is that relationships between neighboring spatial units are much stronger than between distant ones [10]. Spatial dependence can be captured and geo-visualized as spatial varying patterns across the landscape. Methods that can analyze spatial dependence can then be able to calculate how geographically explicit attributes existing in one household or a neighborhood influence or are influenced by those in the neighboring spatial units. However, most empirical studies do not account for the spatial dependency of geographically explicit factors that play an important role in shaping smallholder decision-making processes. The inherent difficulty emanates from the lack of a clear, spatially explicit methodology that can detect and map location-specific spatial dependence. Besides, Wiggins [1] says that comprehensive spatial data disaggregated at the local level to support localized spatial analysis hardly exists. The risk of relying on aggregated spatial data to detect local spatial dependence, according to Nthiwa [25] and Głębocki et al. [10], is that aggregated data masks important underlying local factors and obscure emergent local spatial patterns. The lack of a method to analyze local spatial dependence makes many existing empirical approaches to turn a blind eye to the geographical reality of the spatial context of determinant that influences agriculture production [10,20]. As a consequence, it is difficult for policymakers to design spatially targeted interventions for addressing local level challenges that hinder many resource-poor smallholders, particularly in the marginalized rural areas, from participating in the agribusiness market.

In Geographic Information Systems (GIS), spatial dependency is measured using spatial autocorrelation. In essence, Fotheringham [26], notes that the construction of spatial autocorrelation depends on spatial dependency. He describes spatial autocorrelation as a measure of the strength and direction of spatial dependency, whereby observations at locations closer to each other in geographic

space are also more likely to be similar in attribute (positive spatial autocorrelation) than observations farther apart that tend to have dissimilar attributes (negative spatial autocorrelation).

The continued development of GIS has led to the development of advanced local spatial geostatistical methods that analyze and model local spatial autocorrelation [27]. For example, Global Moran's I spatial geostatistics is commonly used. The method identifies presence of autocorrelation (homogeneous and heterogeneous patterns) in variables at a global 'entire dataset' level. However, according to Ord and Getis [28], the shortcoming of Global Moran's I method only detects spatial autocorrelation at global (entire dataset) and not for disaggregated local-level data. Thus, an additional analysis is required to calculate spatial autocorrelation for a local-level disaggregated data. Cluster and Outlier Analysis (Anselin Local Moran's I) method is a spatial geostatistics method that has been developed to detect the presence of local-level spatial autocorrelation and to map spatial clusters [29]. The method classifies statistically significant *p*-values into Hot spots (High–High clusters, High–Low clusters), Cold spots (Low–High clusters, Low–Low clusters), and non-significant areas. To identify spatially explicit factors causing these spatial clusters, Geographically Weighted Regression (GWR) is used. The GWR identifies statistically significant spatial factors behind the local spatial autocorrelation [29]. In this study, by combining the three methods above, we identified specific localities with statistically significant concentration of high values (hot spots) and concentrations of low values (cold spots), and factors causing these spatial clusters [23,30]. Mapping such localities is important because it would allow policymakers to make evidence-based decisions and also enable them to design appropriate spatially targeted interventions tailored to local contexts. The aim of this paper was thus to map, analyze and geo-visualize geographically explicit determinants, in detecting the presence of statistically significant local spatial patterns in Nyando and Vihiga study areas. This was done to unearth local spatial factors that influence smallholder households' decisions to participate (or not) in agribusiness in the two study areas.

## 2. Materials and Methods

### 2.1. Study Area

Two distinct study areas (Figure 1), Nyando, and Vihiga sub-counties located in Western Kenya were selected for this study.

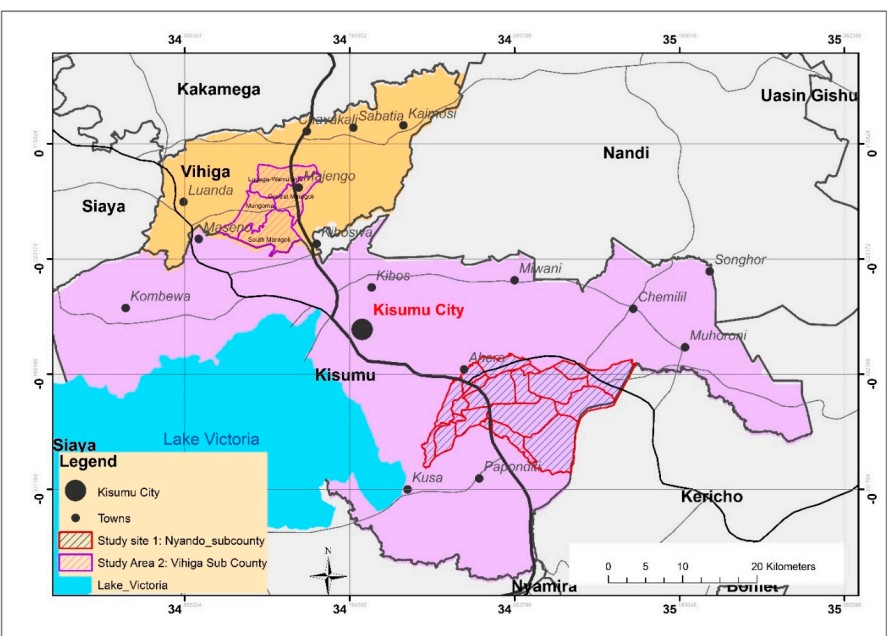

**Figure 1.** Geographical location of Nyando and Vihiga study areas.

The justification for using two geographically distinct study areas in our analysis was to allow for the cross-validation mechanism of our results, and by extension, as a test of the robustness of our spatial analytic method developed. We assumed that the results of local spatial dependence and heterogeneity for one study area could only be considered conclusively robust and reliable if a second study area with distinct characteristics was included in this study. Thus, the results of the two study areas provide a critical reflection on the usability of the method developed in this paper.

The selection of the two study areas was based on several factors, including their high population density, high prevalence of food insecurity, and their agroclimatic and agroecological potential for agricultural production. According to the Kenya National Bureau of Statistics census report, 2019, Vihiga county has the highest population density in Kenya at 1300 persons per $km^2$ against the nation's 92 persons per $km^2$ [31]. Nyando's population density is lower at 400 persons per $km^2$. In terms of land use, both areas are characterized by heterogeneous land-use systems with farming typologies ranging from pure subsistence, mixed subsistence, to cash crop-oriented farming. The main cash crop grown by households, at relatively small farms, include tea and coffee production in Vihiga and sugarcane and rice production in Nyando. Both areas are predominated by smallholder households whose average farm sizes range from 0.1 to 2.0 acres. The two study areas exhibit spatial and biophysical variability in terms of topography, soil types, altitude, and rainfall. They also receive bimodal rainfall, with Vihiga receiving higher amounts than Nyando. The topography of Nyando is predominantly flat while Vihiga's is undulating in the east and gently flat in the west.

## 2.2. Data Collection Methods

A geocoded household survey, using face-to-face interviews and questionnaires was conducted from June to November 2018. We used households as our sampling units and a total of 392 households were interviewed in the two study areas. A questionnaire with closed and open-ended questions was our main survey instrument and was administered to the households with the help of 10 research assistants from Maseno University. The research assistants were selected based on familiarity with the study area, and the ability to speak the local dialect(s). Before fieldwork, the assistants were trained and incorporated from the initial designing of the questionnaire, translating it to local dialects and pretesting it. The questionnaire covered diverse topics and captured data on biophysical, socio-economic, and agroecological aspects of each household.

Given population distribution characteristics and accuracy, we used Cochran [32] formula to calculate the desired sample size as follows:

$$n_0 = \frac{z^2 pq}{e^2} \tag{1}$$

where

$n_0$ = desired sample size if the population is greater than 10,000.
$z^2$ = standard normal deviation at required confidence level (95% or 1.96).
p = the degree of variability 'heterogeneity' of the population (p = 0.5)
q = 1 − p (proportion in the target population)
$e^2$ = desired level of precision

therefore,

$$n_0 = \frac{(1.96)^2 (0.5)(0.5)}{(0.07)^2} = 196 \text{ sample households } (\textit{for one study area}) \tag{2}$$

The study was conducted per the Declaration of Helsinki, and the protocol was approved by Maseno University Ethical Review Committee, (reference number: MSU/DRPI/MUERC/00633/18). All subjects gave their informed consent for inclusion before they participated in the study.

### 2.3. A Geocoded Sampling Design for Household Interviews

In this study, a well-articulated geocoded data collection strategy was designed for guiding the household survey. In step one, we superimposed the administrative polygon of each study area with a grid cell of 100 by 100 m using the create Fishnet grid function of the ArcGIS software. Secondly, we used ArcGIS software to randomly distribute our predetermined household sample size in the gridded study area polygon. In ensuring a spatially distributed data collection will be achieved, we used a rule-based algorithm to distribute the random sample where a minimum distance between any two random sample points was restricted to 50 m. In step two, the randomized household sample points and the study are grids were then converted into Keyhole Markup Language (KML) layers in ArcGIS and then superimposed on the Google Earth browser's high-resolution satellite image. In the last step, we copied these Google Earth KML layers into Geographic Positioning Systems' "GPS Essential App" preloaded in the Android-enabled phones of each research assistant. In the last step, during the fieldwork household survey, we then used the android phones to easily and accurately geolocate the randomized household points for interviews in the study area. The actual household on which each randomized sample point fell on the google' high-resolution satellite image was prioritized for interviewing. Simple random sampling was used to select any household amongst those enclosed by the 100 by 100 m square grid. These steps are illustrated in Figure 2.

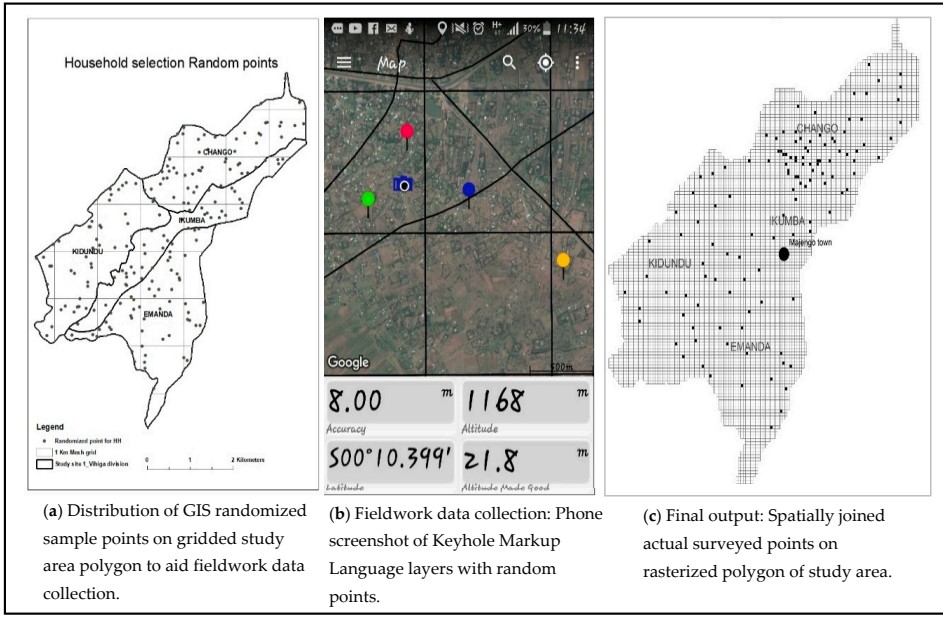

| (a) Distribution of GIS randomized sample points on gridded study area polygon to aid fieldwork data collection. | (b) Fieldwork data collection: Phone screenshot of Keyhole Markup Language layers with random points. | (c) Final output: Spatially joined actual surveyed points on rasterized polygon of study area. |

**Figure 2.** Steps applied in geocoded household survey design; (**a**) distribution of randomized GIS points; (**b**) uploaded KML layers on 'GPS Essential App' in Android phone, and (**c**) Actual surveyed household GPS points.

The quality of data is influenced by the validity and reliability of the method and instruments used to collect data [33]. During fieldwork data collection, data quality management was addressed in various ways. Before and after each day data collection, the principal researcher and the research assistants discussed the data collection formalities, etiquette, emerging issues, and proposed solutions. Additionally, every day we projected and mapped the GPS coordinates points of the administered household questionnaires and uploaded the projected layers in the android phones of the research assistants. This enabled us to identify interview gaps by identifying areas covered and not covered by research assistants.

Before performing spatial geostatistical analysis, the household data were tested for normality, multicollinearity, and goodness of fit using Statistical Package for the Social Sciences (SPSS) software. Subsequently, we used the Exploratory Regression Statistics tool in ArcGIS software to test these variables

for residual spatial autocorrelation, residual normality, and global multicollinearity (of less than VIF < 7.5). Table 1 provides a summary of the descriptive statistics of the variables used in this study.

**Table 1.** Description of data variables used in the analysis.

| Explanatory Variables | Unit of Measure | Variable Description |
|---|---|---|
| Market Participation (Dependent variable) | Binary | 1 if household participate in markets and 0 otherwise |
| **Independent variables** | | |
| **Socio-economic and welfare** | | |
| Gender | Binary | 1 if household head is male 0 otherwise. |
| Education level | Categorical | House head level of education (Primary, Secondary, Tertiary). |
| Family labor availability | Binary | 1 if house head has enough family labor, and 0 otherwise. |
| Family savings | Binary | 1 if head saves money, 0 otherwise. |
| Association membership | Binary | 1 if head belong to a social network and 0 otherwise. |
| Agriculture training | Binary | 1 if the head had training in the last one year, 0 otherwise. |
| **Natural and financial factors** | | |
| Access to agriculture credit | Binary | 1 if the head has access to agric. credit and 0 otherwise. |
| Household assets (USD) | Continuous | The total monetary value of household assets. |
| Livestock assets (USD) | Continuous | Natural Log, the value of livestock assets. |
| Landholding size (acres) | Continuous | Natural Log, landholding size of a household. |
| Land tenure system | Binary | 1 if the head has a title deed and 0 otherwise. |
| Hybrid seeds use and access | Binary | 1 if the head use or has access to hybrid seeds and 0 otherwise. |
| Agriculture extension | Binary | 1 if the head has access to extension services and 0 otherwise. |
| **Biophysical and agroecological** | | |
| Soil fertility level | Categorical | Perceived level of soil fertility (low, medium, high). |
| Slope (derived from altitude) | Ordinal | Household land gradient (flat, gentle, steeply). |
| Impact of pest and diseases | Ordinal | Level of the impact of pests and diseases on crops. (Little or no impact, medium impact, high impact). |
| Impact of climate variability | Ordinal | Effect of drought and famine (low, medium, high) |
| Rainfall adequacy | Ordinal | Level of rainfall (little, medium, high). |
| **Infrastructure and market access** | | |
| Travel time to the market center (Mins) | Categorical | 0–10 min, 11–20 min, 21–30 min, 31 min and above. |
| Travel time to Agrovet shop (Mins) | Categorical | 0–10 min, 11–20 min, 21–30 min, 31 min and above. |
| Distance to the tarmac road (Meters) | Categorical | Proximity to tarmac road by a household. |
| **Institutional factors** | | |
| Market regulations Influence | Ordinal | Perceived level of influence, (little, medium, high). |
| Government policy (subsidy) influence on farming | Ordinal | Perceived level of influence, (little, medium, high). |

*2.4. Modeling Local Spatial Relationships*

In modeling spatial relationships, and in calculating spatial autocorrelation, there are two crucial factors: (1) spatial unit of analysis and (2) territorial distance or spatial unit of analysis [34] that should

explicitly be determined before spatial analysis can be carried out. According to Arsenault, Michel, Berke, Ravel, and Gosselin, [34], choosing the appropriate geographical unit of analysis emanates from the Modifiable Areal Unit Problem (MAUP). Principally, MAUP emanates from, (1) lack of adequate conceptualization, (2) lack of consideration of the scale of measurement, and (3) how spatial data are aggregated or disaggregated [25,34]. The geographical unit of analysis is the extent of a geographic area at which a phenomenon or underlying spatial process occurs [35]. For this study, we used cell grids of 50 by 50 m as our disaggregated geographic unit of analysis. This was achieved by rasterizing the administrative polygon of the study areas by using ArcGIS "create fishnet grid" tool. We then transposed the sampled households' GPS points and their associated attribute data into the rasterized layer to allow cell by cell analysis.

Territorial distance value defines the appropriate spatial unit of analysis. The assumption is that the optimal territorial distance value will be where the underlying processes promoting spatial clustering are most pronounced. According to Getis and Aldstadt, [36], the intensity of spatial clustering is determined by the z-score returned, with the most optimal territorial distance symbolized graphically as the peak z score value. In our analysis, we used the "Incremental Spatial Autocorrelation tool" in ArcGIS to calculate the most optimal statistically significant peak z-scores (Figure 3).

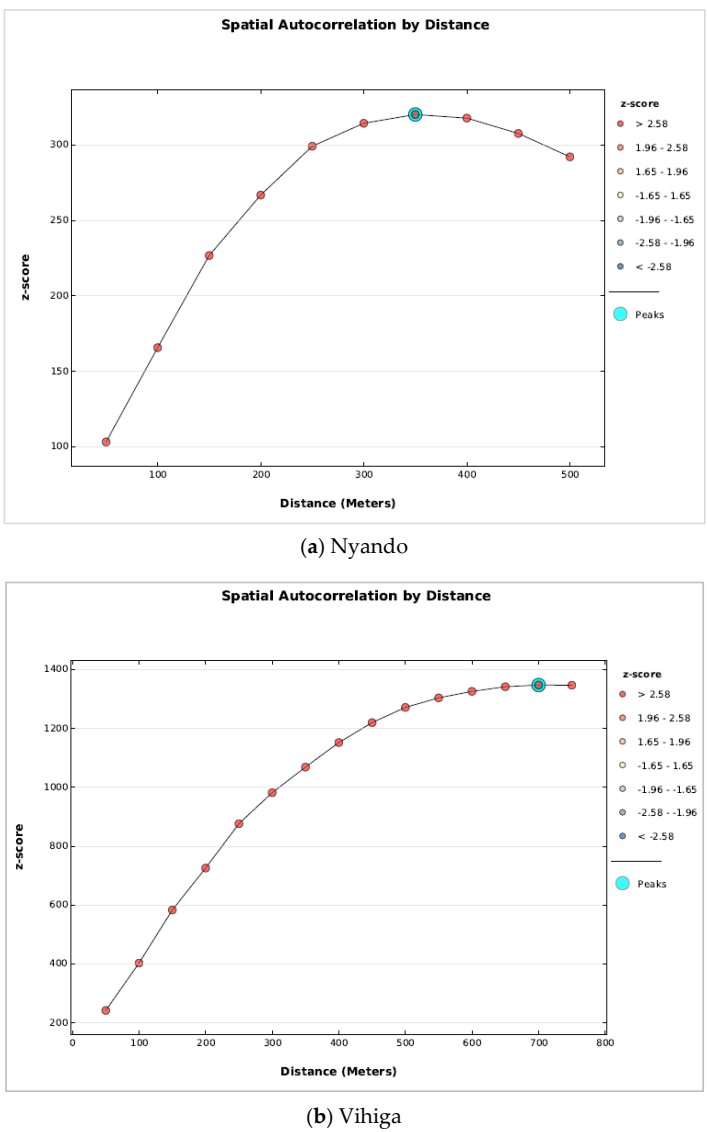

(**a**) Nyando

(**b**) Vihiga

**Figure 3.** Graph showing the most optimal statistically significant peak z-scores of spatial clustering.

Our calculations returned an optimal territorial distance of 350 m for Nyando and 700 m for Vihiga study areas. Subsequently, we used these territorial distances as our input value in Cluster and Outlier Analysis in calculating local spatial autocorrelation and geographically weighted spatial regression analysis in ArcGIS. Spatial analysis was based on rasterized cell grids with their associated attribute data.

To improve the accuracy of spatial regression results and interpretability of the output statistics, two problems associated with modeling spatial relationships should be addressed beforehand. First, Głębocki et al. [10] note that, in reality, the spatial relationships are not homogeneous, meaning that factors promoting spatial autocorrelation have different potential for interactions. In accounting for this shortcoming, we used a row-standardized spatial weight matrix [28]. The spatial weights matrix quantifies the spatial relationships that exist among the features in the dataset and row standardization creates proportional weights to account where certain features may have an unequal number of neighbors [36,37]. It is noted by Getis and Aldstadt [36] that this method is popularly used by different authors as it is effective. The second problem as highlighted by Castro and Singer [38] is that the spatial data from local features can artificially inflate the spatial statistical significance (i.e., type 1 error where one may incorrectly reject the null hypothesis). To account for this shortcoming, we applied False Discovery Rate (FDR) correction that adjust the critical *p*-value thresholds [38] in our Cluster and Outlier Analysis calculations.

## 2.5. Analyzing Local Spatial Autocorrelation

Our null hypothesis was that there is complete spatial randomness of data on households not participating in markets across the two study areas; that is, no spatial pattern of factors that impede smallholders' participation in agribusiness in both study areas. We used three methods to test our hypothesis. In the first step, we used Global Moran's I spatial autocorrelation method to assess the presence or absence of spatial patterns in our dataset. According to Goodchild [35], the method calculates the z-score and *p*-values which indicate whether to reject or accept the null hypothesis. However, Global Moran's I result only reveal spatial autocorrelation's 'spatial patterns' for the entire dataset but not at the local level 'households and their neighbors'. Accordingly, in step 2, we used Cluster and Outliers Analysis (Anselin Local Moran's I) method to detect the presence of local-level spatial patterns and clusters and to determine if these spatial clusters are statistically significant or are resultant from complete spatial randomness of data in the study area. This method categorizes spatial units to have either positive or negative spatial patterns at significance ($p < 0.05$). The output of this method is standard deviations (LMi index, LMiZ score, LMip values) for statistical analysis and geo-visualized map (Gi Bin/CO-Type column) statistics that classify all the statistically significant *p*-values into three types; Hot spots (High–High clusters, High–Low clusters), Cold spots (Low–High clusters, Low–Low clusters) and non-significant areas [23]. The justification for using the Cluster and Outliers Analysis method in our study is that it supports local level spatial analysis and interpretation of results and also supports the use of a spatial weight matrix [28,30].

In the last step, we used Geographically Weighted Regression (GWR) to examine geographically significant local factors that explain households' non-market participation; in other words, factors behind the observed spatial patterns identified in step 2. According to Fotheringham, Brunsdon, and Charlton [29], the GWR model is a non-stationary technique that measures spatially varying inherent relationships for a set of coefficients. Since the variables being estimated vary continuously over the study area, their "surface can be geo-visualized and interrogated for relationship heterogeneity" [29]. In geo-visualizing localities and households where the concentration of spatial factors hindering market participation was most pronounced, we used the predicted probability score of household market participation and geo-visualized standardized residuals from non-market participants households. The findings are presented as inferential spatial statistics and geo-visualized as GIS output maps in the results and discussion section below.

## 3. Results and Discussion

### 3.1. Characteristics of Sampled Household

A total of 392 sample household heads were interviewed comprising 21% aged between 18–35 years, 55% aged between 36–60 years, and 24% aged 61 years and above. The sample size was comprised of an almost equal number of males (49.7%) and females (50.3%). The average household size was 6.9 persons, which was considerably higher than the national average of 3.9 persons per household as per the latest Kenya census report (Kenya Bureau of Statistics, 2019). We observed large family sizes, with households having 5 persons and above comprising 85% of the total sample. This is quite significant in our study as it is a factor that exerts a huge demand for both household food demands and pressure on cultivable land, especially for the next generation. The average landholding size was found to be 2.12 acres though a larger percent (62%) of sampled households landholding sizes was below 2 acres. Subsequently, all these factors could have contributed to higher food insecurity incidences observed. About 49% and 36% of sampled households in Nyando and Vihiga, respectively, reported having experienced food shortage in the last one year. The findings correlate with average food insecurity of 40% for both counties reported in the Kisumu and Vihiga County Integrated Development Plans (2018–2022).

For the context of our study, we considered agribusiness markets participants as those households, regardless of farming production typology and the scale of production, which sell certain quantities of either crops or livestock products, personally or through intermediaries to either informal or formal markets. Non-market participant households were categorized as those who do not sell any farm or animal produce to the markets. The results (Table 2) show that, overall, households market participation is low (31%) in both areas, with a higher percentage (69%) of households in both Nyando and Vihiga not participating in markets.

**Table 2.** Cross-tabulation of the percentage of (a) households' non-market participation, and (b) household farming production orientation.

| | (a) Household Market (non)Participation | | | (b) Farming Production Orientation | | | |
|---|---|---|---|---|---|---|---|
| | Nyando | Vihiga | Overall | **Type** | Nyando | Vihiga | Overall |
| | *Percent* | *Percent* | *Percent* | | *Percent* | *Percent* | *Percent* |
| **No** | 75% (147) | 62% (122) | 69% (269) | Pure subsistence | 20% (40) | 22% (43) | 21% (83) |
| **Yes** | 25% (49) | 38% (74) | 31% (123) | Mixed subsistence | 66% (129) | 47% (93) | 57% (222) |
| | 100% | 100% | 100% | Semi-commercial | 2% (3) | 27% (53) | 14% (56) |
| | | | | Horticulture | 12% (24) | 4% (7) | 8% (31) |

Different farming production orientations were observed in the study areas (Table 2). Overall, a high percentage of households' food production in both the study areas is oriented towards subsistence farming, while only a marginal 14% and 8% were oriented towards semi-commercial and horticulture farming, respectively. The main food crops grown both for food crops and for selling included maize, beans, bananas, vegetables, mangoes, avocado, and pawpaw. Cash crops included coffee and tea in Vihiga and sugarcane and rice in Nyando.

### 3.2. Results of Local Spatial Autocorrelation

The Global Moran's I statistics results (Table 3) revealed the presence of spatial autocorrelation in our data set, with a global Moran's index = 0.713, z-score = 242.3, ($p < 0.000$) for Vihiga and global Moran's index = 0.903, z-score = 383.86 ($p < 0.000$) for Nyando dataset.

**Table 3.** Global Moran's I result indicating the presence of spatial autocorrelation.

| Study Area | Global Moran's Index | Expected Index | Variance | z-Score * | *p*-Value |
| --- | --- | --- | --- | --- | --- |
| Vihiga | 0.713 | −0.000135 | −0.000 | 242.34 | 0.000 |
| Nyando | 0.903 | −0.000029 | 0.000 | 383.86 | 0.000 |

* The high z-score reflects the high intensity of spatial clustering.

Given the statistically significant z-score, there is a less than 1% likelihood that this clustered pattern could be the result of random chance. Thus, we reject our null hypothesis. This confirms that in both study areas, there is presence of spatial clustering and patterns that could not be the result of complete spatial randomness of data.

Results of Cluster and Outliers Analysis (Anselin Local Moran's I) for the study areas show the presence of local spatial autocorrelation. This is presented as statistically significant (>+1.96 > +3.4, $p < 0.05$) local spatial clusters of high values (Hot spots) and clusters of low values (cold spots) geo-visualized in Figures 3 and 4.

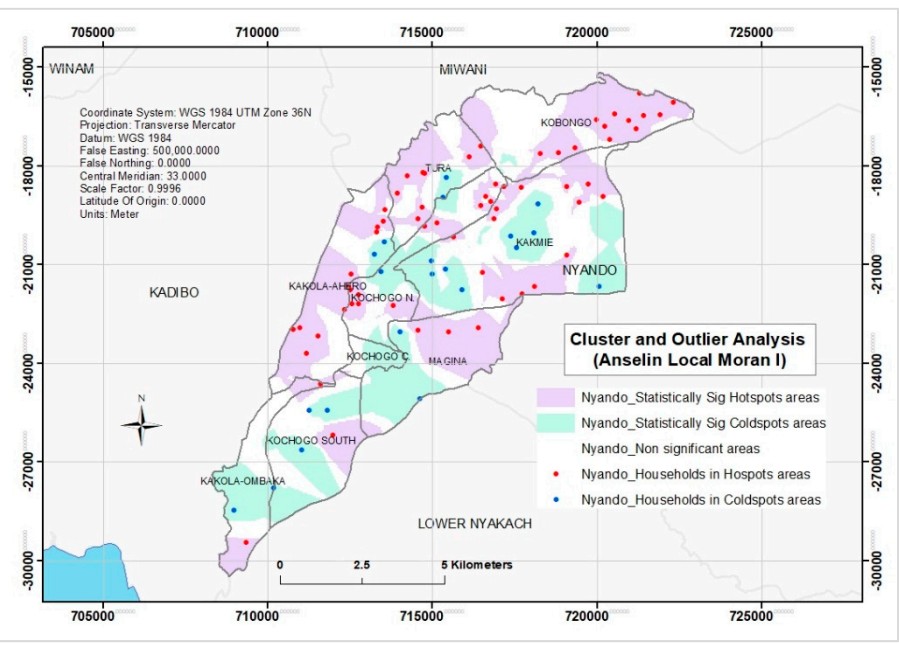

**Figure 4.** Map of Nyando showing local spatial clusters with a higher concentration of poorer households (hot spots) and richer households (cold spots).

In the maps (Figures 4 and 5), the local Moran's I *p*-value, significant at 0.05 using False Discovery Rate (FDR) correction is symbolized as hot spots and cold spot areas in the legend. In both maps, the hot spots and cold spots areas are statistically significant local spatial clusters of high values and low values, respectively. These spatial clusters are surrounded by non-significant areas (white patches). We superimposed the spatial clusters with GPS points of households that did not participate in markets, with red dots being non market participants households in hot spots areas and blue dots showing non market participants households in cold spots areas.

An important observation from the two maps is that factors impeding market participation have several distinct local spatial clustering across the two study areas. The difference in spatial clustering could be explained by the dissimilar social-spatial resources existing in the study areas, and the capability of each household to maximize its livelihood assets to exploit those resources. These maps provide for easier visual interpretation by policymakers for spatial targeting of interventions. The observed spatial patterns are not a result of complete spatial randomness, which means that

underlying spatial explicit factors are causing these spatial clusters. These factors are explained in Section 3.4.

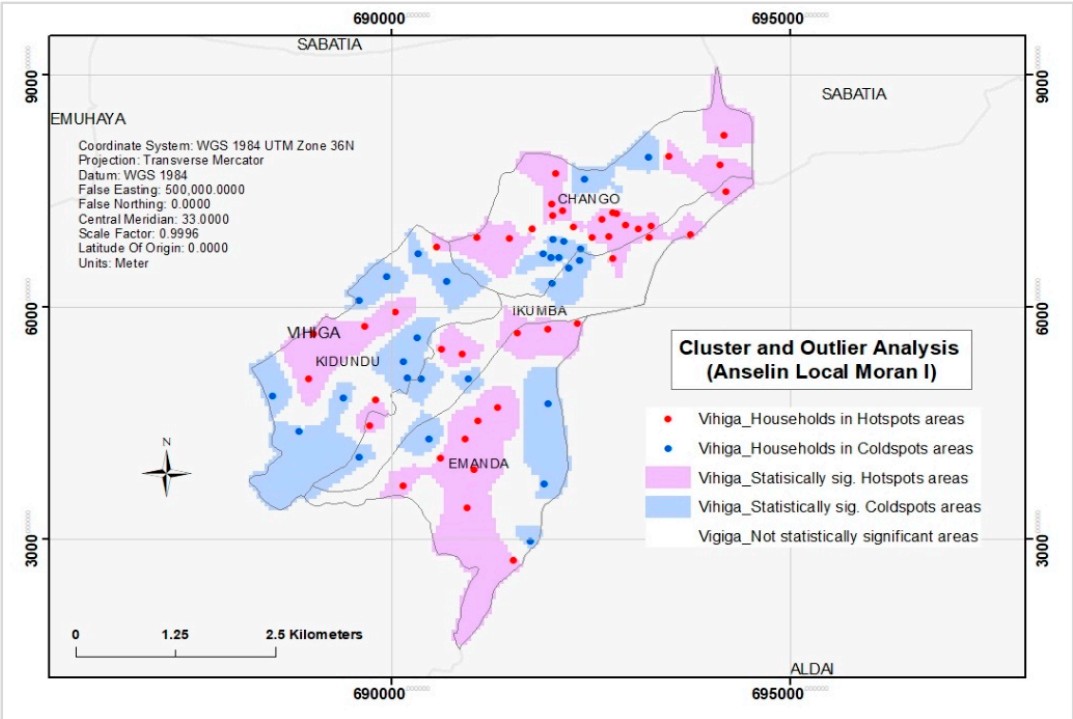

**Figure 5.** Map of Vihiga showing local spatial clusters with a higher concentration of poorer households (hot spots) and richer households (cold spots).

### 3.3. Mapping Local Spatial Complexity of Causative Factors of Non-Market Participation

The results of spatial proximity to supportive infrastructure (Table 4) show that closeness to road, town and water source had minimal influence on poor smallholders' decisions to participate in agribusiness.

**Table 4.** Results of spatial proximity analysis of non-market participants households.

| | Non-Market Participating Households | | | |
| | Nyando | | Vihiga | |
| **Euclidean Distance** | **In Hot Spot (n = 63)** | **In Cold Spot (n = 21)** | **In Hot Spot (n = 43)** | **In Cold Spot (n = 28)** |
|---|---|---|---|---|
| 1 km buffer from tarmac road | 38 (68%) | 4 (19%) | 26 (65%) | 16 (57%) |
| 1 km buffer from main town | 6 (10%) | 0 (0%) | 5 (12%) | 3 (11%) |
| 500 m buffer from river | 38 (22%) | 4 (19%) | 0 | 0 |

Contrary to our expectation, in both Nyando and Vihiga, the majority of poorer households in hot spots areas were close to tarmac roads, town centers, and water sources (river). Yet, these factors are often viewed as positive drivers of agribusiness and market participation. For example, 68% and 65% of poor households in Nyando and Vihiga, respectively, were located within a radius of 1 km from the main tarmac road. Equally, the spatial proximity maps (Figure 6) revealed a high concentration of non-market participants households (both red and blue dots) within a 1-km buffer of towns, tarmac roads, and a 500-m buffer from rivers.

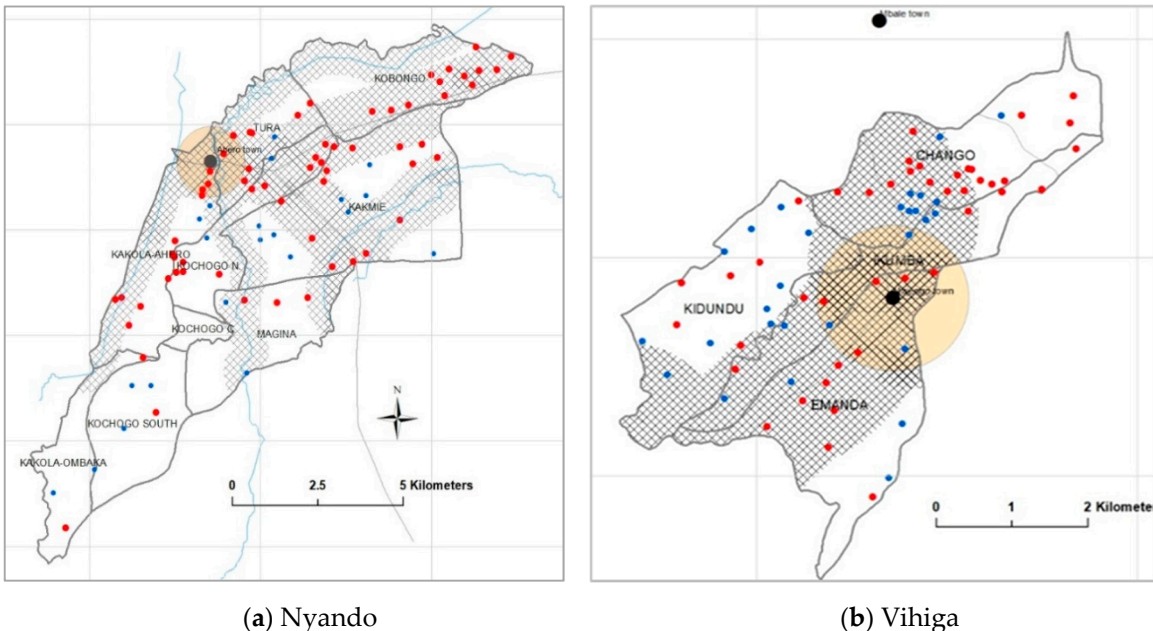

(**a**) Nyando            (**b**) Vihiga

**Figure 6.** The two maps reveals that, in both Nyando and Vihiga areas, there were more poorer households (red dots) than richer households (blue dots) located within a I kilometer buffer (crosshatched areas) from the tarmac road, main town, and rivers.

Such high percentages of poorer people staying close to basic services would imply that these services had little influence on their decisions to participate in markets. Several studies [39,40] postulate that the ability (or inability thereof) of poor households to exploit the opportunities for improving their livelihoods is influenced by their level of poverty and multiple deprivation status. This has a consequence to policy in that improving one aspect of factors that impede agribusiness development cannot produce intended consequences, and hence a holistic approach is need. Pro-poor agricultural development proponents advocate for smallholder agriculture diversification in both farm and non-farm activities as the most promising pathways to accelerate poverty and income inequality reduction [41]. However, these approaches should be accompanied by an integrated and multidisciplinary interventions. and spatial targeting can improve the process.

*3.4. Comparing the Market Participation Odds for Both Richer and Poorer Households in Hot and Cold Spots in Nyando and Vihiga*

The regression results (Table 5) show geographically explicit factors that influence market participation decisions of poor smallholder households in the two study areas. These factors are associated with the spatial clusters of hot spots and cold spots geo-visualized by Nyando and Vihiga maps. The two study areas exhibited both similarity and dissimilarity of spatially explicit factors impeding market participation.

In Nyando, the results show that occupation, education level, household and livestock assets, savings, landholding size, membership to a social group, and travel time to output market were spatially and statistically significant factors impeding poor smallholder participation in agribusiness markets. In Vihiga, the regression results revealed that education level, savings, land size, training, and travel time to markets were statistically significant factors that impeded household market participation.

Landholding size negatively and significantly ($p < 0.05$) influenced the decision of households to participate in markets. *Ceteris paribus,* the odds ratio of the likelihood for smallholders to choose to participate in agribusiness markets decreased by a factor of −1.160 for Vihiga and −1.537 for Nyando, with a unit decrease in land size, at 95 % confidence level. Supportive evidence from the study findings shows that the majority of smallholders owned very small land sizes that were uneconomical to support

surplus production for selling to the markets. Again, they barely produced enough to support their household food demands. The majority of households (58%) said that the food they produced was not enough to sustain them till the next harvest. There is a need for local governments to adopt spatially based integrated planning that promotes pre-emptive co-ordination of different land-use functions and activities as efficiently as possible to maximize the 'benefits' of a given locality [42]. This can be achieved through the adoption of spatially dependent studies that model and predict different spatial scenarios based on optimized resource (re)allocation according to their suitability and availability.

**Table 5.** Spatially and statistically significant factors influencing smallholder participation in agribusiness in Nyando and Vihiga.

| Vihiga Sub-County | | | | | |
|---|---|---|---|---|---|
| **Predictor Variable** | **B.** | **S.E.** | **Wald $X^2$** | **$p$-Value** | **Odds Ratio** |
| Education Level | −2.034 | 1.124 | 3.278 | 0.070 | 0.131 |
| Savings | −1.348 | 0.428 | 9.927 | 0.002 | 0.260 |
| Land size | −1.160 | 0.547 | 4.496 | 0.034 | 0.313 |
| Training | −0.850 | 0.516 | 2.718 | 0.099 | 0.427 |
| Travel time to markets | −1.751 | 0.870 | 4.052 | 0.044 | 0.174 |
| Constant | 3.740 | 1.181 | 10.021 | 0.002 | 42.101 |
| Nagelkerke $R^2$ | 0.33 | | | | |
| −2 Log likelihood | 205.51 | | | | |
| **Nyando Sub-County** | | | | | |
| **Predictor Variable** | **B.** | **S.E.** | **Wald $X^2$** | **$p$-Value** | **Odds Ratio** |
| Occupation | −1.662 | 0.784 | 4.495 | 0.034 | 0.190 |
| Education level | −5.515 | 1.686 | 10.701 | 0.001 | 0.004 |
| Household assets | 2.215 | 1.327 | 2.789 | 0.095 | 9.165 |
| Livestock assets | −1.286 | 0.748 | 2.958 | 0.085 | 0.276 |
| Savings | 0.916 | 0.561 | 2.665 | 0.103 | 2.499 |
| Landholding size | −1.537 | 0.702 | 4.794 | 0.029 | 0.215 |
| Social group member | −1.296 | 0.488 | 7.037 | 0.008 | 0.274 |
| Travel time to market | −2.337 | 0.828 | 7.966 | 0.005 | 0.097 |
| Travel time to Agrovets | 2.119 | 0.846 | 6.280 | 0.012 | 8.323 |
| Constant | 5.037 | 1.227 | 16.850 | 0.000 | 154.080 |
| Nagelkerke $R^2$ | 0.421 | | | | |
| −2 Log likelihood | 154.82 | | | | |

Low level of education has often been reported amongst the key barriers of market participation of poor households [43]. Our findings corroborate this, as we found education to be statistically significant ($p < 0.01$) in influencing household market participation in both study areas. In interpreting the odd ratio, households with low education levels were 0.004 less likely to participate in markets than those with one level higher of education, all other factors kept constant. From a local-scale perspective, non-market participant households with a low level of education (primary school education and below) were significantly higher (51%) in the high-cluster hot spot areas, than those in cold cluster zones (15%). The same scenario was found in Vihiga, where a relatively high percentage (45%) of non-market participants households in hot spot areas had a primary level of education as compared to only 21% of respondents in cold spots areas. This would mean that spatially targeted interventions in terms of education and training in hots spot areas would lead to a relatively higher probability of improving the market participation of those households. Even though education level was found to be a positive determinant of market participation, our household survey findings revealed that agribusiness skills of household head declined as the level of education increased (Figure 7).

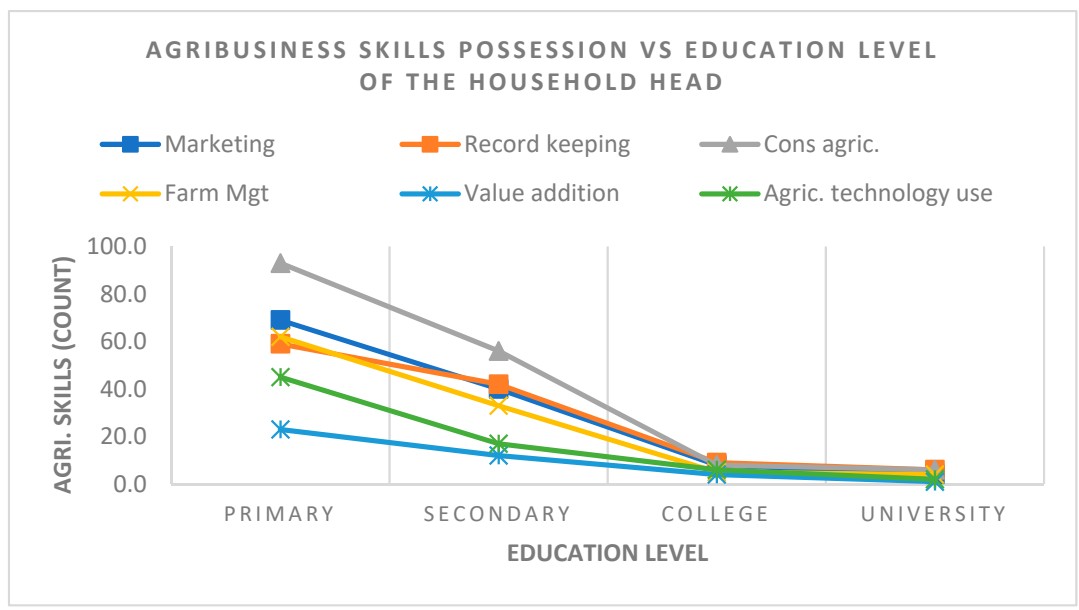

**Figure 7.** Survey results showing households head education level vs agribusiness skills possession.

This implies that in the study area, a higher level of education of the household head did not translate to more agribusiness skills as often presumed. Additionally, our survey results found half (50%) of the household heads with college and university education were in the formal (salaried) employment and not engaged in farming. This raises an important question on whether general education improvement among poor households would be an effective strategy that can improve their farming livelihoods. We argue that rather than spatially blind policy interventions that often advocate for the blanket improvement of general education among poor smallholder, a spatially explicit methodology could be a useful alternative for identifying specific hot spots localities where households have a deficiency in relevant agribusiness skills for spatially targeted interventions.

Household savings was a statistically significant factor that influenced households agribusiness market participation. *Ceteris paribus*, there is 0.266 odds of a household to participate in the market in Nyando if it does not have savings while in Vihiga there is 2.49 odds of a household with savings to participating in the market. From inferential statistics, the majority of households in hot spots areas in both study areas had little savings. In Nyando, 73% of sampled households in hot spots had no savings, while in Vihiga, not a single household in hot spot areas had any savings. Likewise, lack of savings was also higher in households located in cold spot areas, with 67% and 50% of households in cold spot areas in Nyando and Vihiga, respectively, indicating not to have any savings. Lack of monetary savings among the poor households coupled with lack of access to alternative credit sources has been identified in many works of literature as a formidable barrier to poor smallholder's market participation. A pro-poor agricultural policy that promotes a saving culture while enhancing access to affordable credit amongst poor households could empower them to increase their wealth and savings. This in turn would enhance their participation in agribusiness. Spatially targeted analysis for identifying poverty-stricken neighborhoods, locations served and not served by small and microfinance institutions, and identification of most suitable areas for locating these services could generally promote smallholders' credit access and savings culture, factors which are key in promoting agribusiness adoption.

Travel time to output markets was found to be a statistically significant factor that influenced market participation in both study areas. The negative influence indicates that households located farther away from the markets had a higher probability of not participating in markets. All factors kept constant, the odds of a household farther away from the output market to participate is 0.097 in Nyando and 0.174 in Vihiga, as would those near market centers. Of the total households in hot

spots areas, 67% in Vihiga and 49% in Nyando indicated they took 30 minutes and above to access the nearest output market. In the cold spots' areas, 33% of sampled households in Nyando and 36% in Vihiga took 30 minutes and above to access the nearest markets. Spatial targeting could inform policymakers on resource (re)allocation for market infrastructure provision in areas not served by markets to improve poor smallholder market access and participation.

Livestock and household assets ownership were found to significantly influence market participation in Nyando. Low livestock asset endowment was associated with a significantly lower probabilities of households to participate in markets. In interpreting the odds ratio, if all factors are kept constant, there are 0.276 odds in the likelihood of a household not participating in the markets if its lacking livestock assets. In both study areas, results indicate unequal assets ownership between men and women; with women owning more of low-value assets (poultry) and men owing more of higher value assets (cattle and goats). Whereas gender-differentiated assets ownership may emanate from a multitude of reasons including inherent repressive culture and traditions, women assets ownership was found to influence household's food production. Half (49.7 %) of interviewed respondents indicated that women assets ownership in both the study areas had a medium to high influence on household production practices. In some empirical studies [44], livestock assets have often been viewed as liquid assets for poor households in not only reducing risks but also as a buffer to food security, in addition to increasing household wellbeing.

Lack of training in modern agribusiness practices was found to be spatially significant ($p < 0.01$) in limiting smallholder market participation in Vihiga. Only a marginal (17%) of the total sampled households indicated they had received training in the last year. For households located in hot spots areas, only 4% indicated having received training. There is 0.427 odds of a household with no training to participate in markets as would be a household with relevant agribusiness training skills. In improving access to agriculture extension services, local government officials could benefit from spatially dependent study outputs that spatially identify localities with higher clusters of smallholders deficient in certain agribusiness skills.

Accessibility of input sources (Agrovet stores) was found to positively influence market participation in Nyando. In interpreting the odds ratio, *ceteris paribus*, there is 8.323 odds of a household located farther away from an agrovet store to participate in the market than it would be for the same household if it is near the input market. In Vihiga, 67% of households in hot spots areas took 20 minutes and above to walk to the nearest agrovet store, while 71% of those in cold spots took 20 minutes and above to walk to the nearest agrovet store. Ease of access to supportive agriculture infrastructure and services has been shown to improve farmers' market participation [45], especially in rural areas. While it is difficult to map and geo-visualize the location of these services using theoretical studies, spatially dependent analytic approaches, and outputs become indispensable.

## 4. Relevance of the Spatially Explicit Research Outputs in Improving Spatial Targeting of Intervention and Policy

A common concern for empirical studies is the generalizability and replicability of their study findings to a broader context in informing public policy. Principally, spatially explicit studies highly depend on the quality of spatial data and clarity of methods used in their analysis. The axiom "garbage in garbage out", is also applicable in GIS-based spatial data analysis. This implies that the quality of spatial data used, and by extension, the study design used to collect it, should be among the most important considerations for researchers if the study outputs are to be relied upon in informing policy and being able to be reproducible elsewhere. We postulate that spatially dependent empirical studies should address these concerns by designing a well-articulated geocoded data collection strategy. For our study, the design and application of the geocoded household survey strategy enabled us to collect quality geocoded data and the application of local spatial analytic methods addressed the concern of relevancy and informative output. Equally, the quality of our household survey was enhanced by incorporating web-based geospatial tools that helped us to easily and accurately geolocate

sample households in collecting georeferenced data. The spatially explicit methodological approach provided in this study could enable other researchers to replicate the study elsewhere.

Another concern for empirical studies is the generalizability of the study findings. It can be argued that empirical findings may only be valid for a narrow time-scope. This argument emanates from the fact that geocoded surveys capture point data for that particular moment in time, thus localizing the findings and interpretations thereof by binding them in both space and time. However, in principle, territories exhibit spatially heterogeneous characteristics due to the diversity of local geographic specificities and levels of territorial capitals endowment. Even though these territorial characteristics are dynamic, they rarely change rapidly, enabling projection of study findings even when such studies are based on such local geographic parameters. Thus, the correct choice of geographic explicit variables and their analysis can be used to inform decisions in both short- and long-term planning scenarios. While recognizing the multiplicity of parameter variables that can be used to capture local spatial autocorrelation, the spatial unit of analysis used, and the level of disaggregation of spatial data used should be a key consideration in studies that analyze local spatial relationships. As such, the broader relevance of this study would be pegged on the applicability of all these factors discussed, as well as the ability of other researchers, and local governments to apply them in modeling complex local problems.

The application of studies designed to provide local solutions to socio-spatial problems using GIS-based approaches [46–49] has increasingly gained prominence in the recent past. While the capability of local authorities, especially in Sub-Saharan Africa, to apply spatially explicit methodologies in decision-making processes has been questioned, recent developments could promote their relevance and adoption. First, most of the policy directives in Kenya, and Sub-Saharan Africa in general, are often formulated at the national level, but their implementation, is carried out at the grassroots level by relevant local authorities. Secondly, social problems have increasing become interwoven in socio-spatial complexity and their manifestation is most evident at the local level, rather than a regional or national level. In light of this, local governments are embracing Geographic Information and Communication Technologies (or "Geo-ICT") and Spatial Decision Support Systems (SDSS) in their day-to-day problem-solving and decision-making processes to address complex social-spatial problems.

The use and adoption of the spatially explicit methodologies, like the one espoused in this study, for analyzing geographically referenced data are bound to increase in the future due to the increasing accessibility and affordability of Geo-ICT and SDSS tools and methods. Buoyed by the easily accessible and web-based geographic data and information, a plethora of Geo-ICT tools, and SDSS analytic techniques continue to be developed and embraced for deconstructing geographic complexity. Additionally, a multiplicity of geospatial based courses, and open-source software has also been developed by higher learning institutions to build the capacity of both local government staff and new students. In Kenya for example, the use of GIS as SDSS for spatial thinking and planning in County governments has been anchored in law through County Governments Act (No. 17 of 2012) and Urban Areas and Cities Act (No. 13 of 2011). These Acts provide both legal and policy framework for institutionalization and adoption of GIS in County governments. Under these Acts, County governments, including Vihiga and Kisumu counties used in this study, are mandated to develop County Integrated Development Plan (CIDP)_ and establish GIS databases to support County spatial planning. All these developments has increased the demand for GIS courses and professionals, and several universities in Kenya and elsewhere have rolled out courses in this domain to bridge the demand gap.

## 5. Conclusions

Studies that use spatial decision support systems in analyzing socio-spatial complexity in unearthing barriers and facilitators of market participation may provide a better-contextualized understanding of local dynamic forces and interactions that influence smallholder agriculture systems. This paper underscores the importance of designing spatially targeted interventions that are embedded in the local reality and informed by locally expressed needs of smallholder households.

Geo-spatial analysis using disaggregated local-level data is likely to unearth geographic explicit local factors that impede smallholder from participation in agribusiness markets than would spatially aggregated data analyzed at a higher spatial level. Using Global Moran's I, results have revealed the presence of spatial patterns in our dataset that was not caused by spatial randomness of data. Furthermore, the Anselin Local Moran's I, identified statistically significant local spatial clusters of factors that hinder smallholder participation. Finally, the geographically weighted regression identified spatially significant causative factors impeding market participation in the study areas. The results show that occupation, education level, livestock assets, savings, landholding size, membership to social group, training, and travel time to output markets were spatially and statistically significant factors impeding smallholder market participation. Non-market participation was found to result from multifactorial causation linked to the local context. The results have shown that factors hindering market participation were heterogeneous within and across farms and neighborhoods in the study areas. We also found that geographically explicit factors causing non-market participation differs between the two study areas. The geo-visualized regression probability maps are important decision-making visual tools for policymakers to easily identify localities with a high probability of spatial clustering of social-spatial problems of deprivation and inequality. The study has demonstrated how spatially explicit analysis conducted at the local level could help in identifying deprived areas where the most vulnerable, most impoverished and resource-poor households reside.

In designing spatially targeted interventions, policymakers should take cognizance of complex interactions of socio-spatial processes at the local landscape and interrogate how they interact to influence smallholder's decision making and choices. For spatial interventions to be successful, all the factors behind the spatial clustering observed in a locality should be addressed concurrently at the design stage of spatially targeted intervention. Targeting a single factor may fail to enhance smallholder's market participation, since lack of market participation emanates from the complex interaction of multiple factors, as the study has shown. A further study could use a similar approach but carry out analysis at higher spatial units to test explicitly the effect of spatial scale on the patterns of spatial associations which can further help to unearth socio-spatial clustering households multiple deprivations.

**Author Contributions:** Mwehe Mathenge, a doctoral candidate, is the main author. He contributed to the conception, study design, planning, data collection, analysis, and interpretation and prepared a first draft of the paper. Jacqueline E. W. Broerse is the promoter and Ben G. J. S. Sonneveld, the co-promoter of the Ph.D. candidate. They both contributed to the study design and interpretation of results. They also critically revised the paper and approved the final manuscript for submission. All authors have read and agreed to the published version of the manuscript.

**Funding:** This research was funded within the framework of the SPADE project (NICHE-KEN-284) sponsored by NUFFIC and managed by Vrije University Amsterdam. The Spatial Planning and Agribusiness Development (SPADE) project is a collaboration project between Maseno University Kenya, and Vrije Universiteit Amsterdam, implemented in Kenya.

**Acknowledgments:** The authors wish to acknowledge the staff at Kisumu and Vihiga County's department of agriculture, livestock, fisheries, and irrigation for their assistance during field research. We also wish to acknowledge the support given by Maseno university, research assistants for their immense help in fieldwork data collection and all the households participants who provided valuable information during fieldwork.

**Conflicts of Interest:** The authors declare no conflict of interest.

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
