# Peer review of "A Spatially Explicit Approach for Targeting Resource-Poor Smallholders to Improve Their Participation in Agribusiness: A Case of Nyando and Vihiga County in Western Kenya"

_ijgi, doi:10.3390/ijgi9100612_

Round 1

Reviewer 1 Report

This is an interesting paper which aims to analyse the influence of geographic factors on farming decisions of smallholder households. The Authors apply Global as well as Local Moran's I in combination with Geographically Weighted Regression (GWR) to assess the extent in which geographically explicit attributes existing in one household or a neighborhood influence or are influenced by those in the neighboring spatial units. For this purpose, the ArcGIS Spatial Statistics toolbox Cluster and Outlier Analysis Anselin Local Moran's I tool with False Discovery Rate correction was used. The tool classifies statistically significant p-values into Hot spots (High-High clusters, High-Low clusters), Cold spots (Low-High clusters, Low-Low clusters) and Non-significant areas. Identified clusters were then used as input for GWR, with the intent of discovering the spatial factors causing the observed spatial patterns.

The article is well organized, and the research results are clearly presented. Aside from certain language problems, my main issues are with Figure 3, which I believe should be removed and the contents of its caption integrated in the article text, and the contents of section 3.4, which I believe should be expanded, particularly in the “Household savings” paragraph. For more information see the detailed recommendations below.

Detailed recommendations:

Lines 118 and 119: formatting of “km2” should be corrected.

Lines 155-156: Unfinished sentence.

Line 156: “problem” is unnecessary here.

Line 156: did you mean “emanates”?

Line 241: missing “which”?

Figure 3: I believe this figure should be removed as it shows the generic image produced by Cluster and Outliers Analysis, and is not even referenced in the article text. The contents of the figure caption should be integrated into the main article text.

Line 294: Figures 4 and 5 are referenced as being “below” the text, but in reality they have been placed above the paragraph.

Lines 352-356: This section is relatively short, and in contrast to other sections it compares the odds of market participation of poorer households in Nyando to those of richer households in Vihiga. I believe a more thorough analysis, comparing the market participation odds for both richer and poorer households in hot and cold spots in both Nyando and Vihiga should be provided.

Author Response

Dear reviewer,

We are indebted by the very insightful comments and suggestions you have given us in improving our submitted manuscript. We thank you for the time to review our manuscript.  Your comments have improved our manuscript and enriched its contents. We believe that the article and knowledge therein are now better packaged and that it is more coherent, understandable, and relevant to the readers and subscribers of the ISPRS Int. J. of Geo-Information.

In the table below, please find the detailed explanations of the changes we have made to the article and the relevant line numbers the changes have been made. In addition, you will find the tracked changes in the main article for your easier reference.

With best regards,

The authors.

Reviewer 2 Report

Spatial patterns and spatial dependence –
The authors use multiple concepts of space without a clear definition of what they are talking about. Adding the concepts of space, you are using helps the reader. Then describe the problem. The reader is lost in these concepts, until line 82. I see the definition - line 85, but it should be early to help the reader. My comments per line include some conceptual problems when the authors wrote. I believe it will not be too much work to have them fixed.

L44. Join the aims of the study - line 104. It will be more organized the idea.

L. 56 – Space scale or spatial unit. The lowest scale should be a number.
L. 62. Remove etc.
L.67. This sentence is difficult to parse. I would suggest to rewrite it. Also, explain autocorrelation in a positive and negative effect on the variables.
Line 72-73. There is a need for a better definition of spatial dependence and spatial variability because spatial dependence seems to be spatial autocorrelation. If this is the case, you need to talk about negative dependence, positive dependence, or independence.

L 101. Space before parenthesis
L 110. Space before parenthesis

L 121. This journal is a geo-information journal. The authors need to use the concept of scale as a mapping scale. When you say inline 122 small scales, try to change the term as subsistence or the farm area, as you do inline 123.
It will be much better if you write something like: … relatively small farms, ranging from 01. To 0.2 acres, including tea and coffee production in Vihiga… than use scale.

Line 129-131. The mixed model approach is a type of statistical test. What do you mean you used to collect data? Starting at line 132 method is better explained, so remove the first sentence.

L 144. You used binary logistic regression and not a mixed model. I don’t understand why you used logistic regression and then exploratory regression.

Table 1. The authors need to show what these numbers mean and how they were calculated. Explain all the categorical data per variable.

L. 244. Use Table 2 instead of 1a. This table is number 2 and then 2a. And 2b. Table 1 is the descriptive statistical table. The following tables need correction like Table 3 and so on.

L. 298. You don’t’ have table 4 and are already using table 5. Please recheck all the numbers of your tables.

Author Response

(The authors gave the same response as above.)

Reviewer 3 Report

This is a clearly written paper to analyse with the GIS methods the spatial distributions of households and their access and participation into market. A survey/interview is behid the data and various statistical techniques were used to analyse it. I don't feel competent to scrutinize all details of the used techniques but they seem to be applied correctly.

Mr suggestions for the revisions are the following: 1) there is needed to re-write the figure 3 and 4 titles. They are not understandable without reading the main text of the article; 2) there is needed a more critical reflection about the usability of the suggested methods: (a) the collect the representative data is rather laborous, (b) the data points only on one moment, i.e. the changes both inthe population composition of the studied area and the external economy might mean that the interpretations are valid only for a narrow time-scope and not really valuable for long-term planning, (c) it is not sure how cabable the local authorities / universities are in conducting such studies and methods without continuous external funding and training of such research staff, (d) while the methods (seem to) indicate what background variable hint to more or less access to the markets, the situation is in practice much more complicate and depend also on internal family issues, economic and political circumstances, etc. Therefore the suggestion to enhance the education levels of the respondent might lead onto a zero-sum game when all respondents have equally high education levels, so that would not anymore impact teh market access, (e) very important is to consider, are all these empirical results valid only in very specific moment in time and specific location, or are there any chances to utilize these more braodlly uin other context (= is there any braoder relevance of this study empirical results, (f) also suggestions how to conduct properly the empirical data is needed (=if the data is not corrent, then the analysis is not anymore meaningful), (g) reflect also something on power issues as regards market / self-subsistency. You could also tell a bit more how in practice the data was collected.

Author Response

(The authors gave the same response as above.)

Round 2

Reviewer 3 Report

Please, check still the consistency of the liost of references.